# Assessing Interactions between *PNPLA3* and Dietary Intake on Liver Steatosis in Mexican-Origin Adults

**DOI:** 10.3390/ijerph18137055

**Published:** 2021-07-01

**Authors:** Kristin E. Morrill, Victoria L. Bland, Yann C. Klimentidis, Melanie D. Hingle, Cynthia A. Thomson, David O. Garcia

**Affiliations:** 1University of Arizona Cancer Center, Tucson, AZ 85719, USA; 2Department of Nutritional Sciences, College of Agriculture & Life Sciences, University of Arizona, Tucson, AZ 85719, USA; victoriabland@email.arizona.edu (V.L.B.); hinglem@email.arizona.edu (M.D.H.); 3Department of Epidemiology and Biostatistics, Mel and Enid Zuckerman College of Public Health, University of Arizona, Tucson, AZ 85724, USA; yann@email.arizona.edu; 4Department of Health Promotion Sciences, Mel and Enid Zuckerman College of Public Health, University of Arizona, Tucson, AZ 85724, USA; cthomson@email.arizona.edu (C.A.T.); davidogarcia@email.arizona.edu (D.O.G.)

**Keywords:** Hispanic, Mexican-origin, nutrigenetics, nonalcoholic fatty liver disease, NAFLD, PNPLA3

## Abstract

Mexican-origin (MO) adults have among the highest rates of nonalcoholic fatty liver disease (NAFLD) placing them at increased risk of liver cancer. Evidence suggests that a single nucleotide polymorphism (SNP) in the *PNPLA3* gene, rs738409, increases the risk and progression of NAFLD and may modify the relationship between certain dietary factors and liver steatosis. The purpose of this study was to identify whether interactions exist between specific dietary factors and rs738409 genotype status among MO adults in relation to levels of liver steatosis. We analyzed cross-sectional data from a sample of 288 MO adults. Participants completed at least two 24-h dietary recalls. Multiple linear regression was performed assuming an additive genetic model to test the main effects of several dietary variables on levels of hepatic steatosis, adjusting for covariates. To test for effect modification, the product of the genotype and the dietary variable was included as a covariate in the model. No significant association between dietary intake and level of hepatic steatosis was observed, nor any significant gene-diet interactions. Our findings suggest that dietary intake may have the same magnitude of protective or deleterious effect even among MO adults with high genetic risk for NAFLD and NAFLD progression.

## 1. Introduction

Nonalcoholic fatty liver disease (NAFLD) is the most common cause of chronic liver disease in the United States (U.S.) [1]. Increasing rates of NAFLD have paralleled the rising rates of obesity and type 2 diabetes [1]. National estimates of NAFLD prevalence indicate that Hispanic/Latino sub-groups suffer disproportionately to non-Hispanic populations, with adults of Mexican origin having the highest rates compared to other Hispanic/Latino subgroups, including Hispanics of Dominican and Puerto Rican origin [2,3]. For the majority of individuals, NAFLD will manifest as simple nonalcoholic fatty liver; however, for some, NAFLD will progress into a more severe form called nonalcoholic steatohepatitis (NASH). If left untreated, NASH can sequentially progress to fibrosis, cirrhosis, and hepatocellular carcinoma (HCC) [4,5,6,7]. Similar disparities are observed in HCC where incidence in Hispanics is double that in non-Hispanic Whites (NHWs) [8]. HCC remains one of the most fatal cancers with a 5-year cause-specific survival for Hispanics of 20% and 22% for men and women, respectively [8]. Currently, NAFLD is the fastest growing indication for liver transplantation in the U.S. [9] raising further concerns related to access to life-saving medical procedures and care for Hispanics/Latinos [10]. 

Genetics likely play a role in explaining differences in NAFLD prevalence observed among various Hispanic/Latino subgroups [2,11]. Even after controlling for a range of demographic (e.g., age, sex) and clinical (e.g., body mass index (BMI), waist circumference, hypertension, serum lipids, insulin resistance) risk factors, rates of NAFLD in Mexican-origin (MO) adults were found to be significantly higher than participants of Dominican and Puerto Rican origin [2]. One single nucleotide polymorphism (SNP) in the *patatin-like phospholipase domain containing 3* (*PNPLA3)* gene, rs738409, has consistently been associated with increased risk of NAFLD and HCC [12,13]. The SNP represents an Ile148Met substitution (C > G) in the *PNPLA3* gene, a gene that encodes a protein involved in the export of triglycerides out of the liver [14]. In the first study to identify this variant, Romeo et al. (2008) showed those carrying two copies of the risk allele displayed levels of hepatic fat content that were >2-fold higher than non-carriers [12]. In the U.S., the variant is more common among Hispanic individuals with a minor allele frequency (MAF) of 49% compared to 23% among EuropeanAmericans and 17% among AfricanAmericans [12]. The frequency of the variant is found to be particularly high among Mexicans residing in Mexico. In a 2017 study by Martinez et al., the MAF of the SNP was found to be 77% among 211 Mexican adult patients diagnosed with NAFLD [13]. In the same study, patients carrying two copies of the risk allele showed 3.8 times higher risk of having NASH (*p* < 0.05, CI 95%: 3.03–4.79) and 2.32 times higher risk of fibrosis (*p* < 0.05, 95% CI 1.77–3.23), conditions associated with hepatic fat deposition [13].

Current guidelines for the treatment of NAFLD recommend weight loss, physical activity, and diet modification [7,15]. While the American Association for the Study of Liver Diseases (AASLD) does not currently recommend a specific macronutrient composition for NAFLD management, reducing carbohydrate intake (particularly fructose) and increasing omega-3 polyunsaturated fatty acid (PUFA) and monounsaturated fatty acid (MUFA) intake have been associated with improvements in NAFLD outcomes independent of weight loss [16]. While the relationship between fructose intake and NAFLD risk is not fully understood [17], increased intake of omega-3 fatty acids has been found to reduce liver steatosis by increasing fatty acid oxidation and reducing lipogenesis [18]. Interestingly, genetic factors may play a critical role in modifying the effect of dietary carbohydrate and fat intake on NAFLD severity and progression [19,20,21,22].

The rs738409 C>G polymorphism confers susceptibility to NAFLD by enabling the accumulation of PNPLA3 protein at the surface of hepatic lipid droplets, thereby inhibiting the activity of local lipases and impairing the mobilization of triglycerides from these droplets [23]. Given the transcriptional up-regulation of *PNPLA3* that occurs during carbohydrate loading, it is thought that high dietary sugar consumption may exacerbate the effect of the variant by causing an even greater increase in PNPLA3 protein and consequently, higher triglyceride accumulation in the liver [20]. Additionally, carriers of this variant seem to benefit more than non-carriers from hypocaloric, low-carbohydrate diets, despite similar weight loss [22] while seeming to benefit less from supplementation with omega-3 fatty acids [24]. These studies highlight the potential for the variant to modify treatment response to therapeutic NAFLD interventions. Identifying interactions between the variant and specific dietary factors offers an opportunity to understand in what way carriers may respond differently compared to non-carriers to various diet interventions for NAFLD treatment.

Very few studies have investigated interactions between *PNPLA3* genotype status and dietary intake in relation to NAFLD. In a cross-sectional study among 127 Caucasian, African American, and Hispanic children and adolescents, a significant interaction was observed between *PNPLA3* genotype status and omega-6/omega-3 PUFA intake in relation to liver fat content, such that a significant association between omega-6/omega-3 PUFA ratio and liver fat was only observed among individuals with two rs738409 minor alleles (GG) [21]. *PNPLA3* genotype status has also been found to significantly modify the association between total carbohydrate, as well as total sugar intake, and levels of liver fat in a sample of 153 Hispanic children and adolescents [19]. To date, few studies have attempted to characterize the interactions between *PNPLA3* genotype status and dietary intake in adult populations (for exceptions see Scorletti et al., 2015 [24], Stojkovic et al., 2014 [25], and Liangpunsakul et al., 2017 [26]); however, to our knowledge, interactions have yet to be investigated in MO adults specifically, despite their higher risk for NAFLD. Additionally, to our knowledge, no studies have attempted to explore interactions between this variant and dietary intake in relation to levels of hepatic steatosis as assessed by non-invasive transient elastography (Fibroscan^®^), an assessment method widely used for its high-accuracy and utility in guiding clinical management strategies in patients with liver disease [27]. The current study seeks to address these gaps. The purpose of the current study was to determine if the rs738409 genotype modified the relationship of specific dietary factors, including total carbohydrate, total sugar, added sugar, fructose, and omega-6/omega-3 PUFAs, with liver steatosis, in a sample of MO adults with overweight or obesity (n = 288). Based on the findings of previous research that included Hispanic participants, our hypothesis is that gene-diet interactions exist between dietary carbohydrate intake and *PNPLA3* genotype status such that the association of a high-carbohydrate diet with levels of hepatic steatosis is stronger among adults carrying one or two risk (G) alleles.

## 2. Materials and Methods

### 2.1. Study Participation

The current analysis utilized participant data from a previous cross-sectional study. As part of this study, participants completed an in-person, 45–60 min study visit at a clinic specializing in liver health. During the study visit, participants provided a buccal swab sample and completed demographic, behavioral, and psychosocial questionnaires, as well as a brief physical assessment (including anthropometrics), and liver transient elastography (Fibroscan^®^) (EchoSens, Paris, France). Participants were recruited from community-based locations frequented by MO community members such as churches, outdoor markets, community events, and health fairs, as well as community email listservs. Recruitment and study visits occurred between May 2019 and March 2020. Participants provided informed consent to participate and all study procedures were approved by the University of Arizona Institutional Review Board (IRB #1902380787).

Stringent eligibility criteria for the cross-sectional study were established. Participants were required to: (1) self-identify as MO; (2) be 18–64 years of age; (3) have a measured BMI ≥ 25 kg/m^2^; (4) be able to provide informed consent; and (5) have the ability to speak, read, and write in English and/or Spanish. Given that estimates of NAFLD increase with increasing BMI, only individuals with overweight or obesity were recruited. Individuals were excluded if they (1) reported uncontrolled high blood pressure or type 2 diabetes; (2) reported ongoing or recent alcohol consumption (≥21 standard drinks on average per week for men and ≥14 standard drinks on average per week for women) [7]; (3) were taking any medication or supplement known to affect body composition or had a history of exposure to hepatotoxic drugs; (4) had any syndrome or disease known to affect body composition or fat distribution; (5) participated in any structured exercise, nutrition, or weight-loss program within 6 months of recruitment; (6) previously had bariatric surgery; (7) were currently pregnant or breastfeeding; (8) were previously diagnosed with liver disease (including fatty liver disease) or liver cancer, and/or (9) had an active, chronic gastrointestinal disorder (e.g., inflammatory bowel disease, ulcerative colitis, Chron’s disease, celiac disease) that could have influenced usual dietary intake

### 2.2. Study Measures

#### 2.2.1. Anthropometrics

Height, weight, and waist circumference were assessed during the study visit using standardized protocols [28]. Body weight was measured with the participant in street clothes, without shoes, on a calibrated scale to the nearest 0.1 kg (Tanita WB-100A). Height was measured to the nearest 0.1 cm using a stadiometer. BMI was subsequently calculated as weight divided by height squared (kg/m^2^). Waist circumference was measured in the horizontal plane directly at the umbilicus using a Gulick measuring tape recorded to the nearest 0.1 cm. Two measurements were taken for waist circumference unless measurements differed by more than 2.0 cm, in which case a third measurement was taken. The average of the two measurements closest to each other was recorded for data collection.

#### 2.2.2. Liver Steatosis

Transient elastography (FibroScan^®^) was used to measure liver steatosis based on controlled attenuation parameters (CAP) scores and liver stiffness in kilopascals (kPa). Possible CAP values for liver steatosis ranged from 100 to 400 with higher values indicating higher levels of liver steatosis. In the literature, various cut-offs for CAP scores have been proposed to indicate presence and severity of liver steatosis with values above 250 dB/m consistently shown to indicate at least moderate steatosis [29]. This method provides non-invasive, fast, reliable, and reproducible measures of liver steatosis and stiffness (used as a surrogate of liver fibrosis) with good intra- and interobserver levels of agreement [30]. Trained physicians or technicians performed all liver ultrasounds. All participants were fasted for at least three hours before their study visit.

#### 2.2.3. Dietary Assessment

Dietary intake was assessed by three 24-h dietary recalls on 2 weekdays and 1 weekend day within 1–2 weeks after the liver scan. All recalls were administered by the University of Arizona Cancer Center Behavioral Measurement and Interventions Shared Resource (BMISR) via telephone. Trained bilingual dietary technicians, many of whom were MO, conducted all recalls in each participant’s preferred language using the United States Department of Agriculture (USDA) Automated Multiple-Pass Method [31]. The USDA Automated Multiple Pass Method is a five-step dietary assessment method previously shown to be effective in assessing mean energy intake within 10% of mean actual intake in a sample of women with BMI ranging from 20–45 kg/m^2^ [31]. A visual serving size guides (food amounts booklets) was provided to all participants to assist in estimating portion sizes. When possible, dietary technicians had participants refer to this guide during recalls. Additional information regarding usual intake (i.e., “reflective of normal intake”, “more than normal”, “less than normal”) and any reason for unusual intake (e.g., special occasion, ate a buffet) was also collected. To reflect the marketplace throughout the study, dietary intake data were collected using Nutrition Data System for Research (NDSR) software versions 2018 and 2019, developed by the Nutrition Coordinating Center (NCC), University of Minnesota, Minneapolis, MN, USA [32]. Final calculations were completed using NDSR version 2018, (31 July 2020) [32]. The NDSR time-related database updates analytic data while maintaining nutrient profiles true to the version used for data collection. Information on key dietary variables of interest, including total calories, % total calories from nutrients (protein, fat, omega-3 PUFAs, omega-6 PUFAs, carbohydrates, fructose, total sugar, and added sugar), omega-6/omega-3 PUFAs, intakes (g) of dietary fiber, insoluble fiber, and soluble fiber, and glycemic index and load were obtained from the NDS-R software and averaged to obtain an estimate of usual intake. In order to estimate average or usual intake, only participants who completed at least two 24-h dietary recalls were included in the analyses.

To assess plausibility of caloric intake, we assessed the distribution of the residuals of a linear regression of body weight and average caloric intake and intended to exclude participants with residuals > 3SD from the residual mean. A similar strategy for assessing caloric plausibility was utilized by Davis et al. (2010) [19]. No participants met this criterion; therefore, all 288 participants were included in the final analyses.

Given the fact that dietary recalls took place 1–2 weeks after the in-person study visit where participants had the option to review transient elastography (Fibroscan^®^) results with a liver specialist, there was a possibility that participants may have modified their diet based on recommendations made by the specialist (e.g., increase physical activity, lose weight). This would mean that the dietary recall data collected from these participants was potentially not reflective of usual intake. To account for this possibility, participant comments received during the 24-h dietary recalls were reviewed by two authors (K.M. and V.B.). K.M. and V.B. independently reviewed and coded participant comments to determine whether comments were reflective of day-to-day fluctuations in diet (e.g., “not feeling well”, “birthday”, “busy”, and “special occasion”) or long-term diet changes intended to produce weight loss consistent with the liver specialist’s recommendations (e.g., “trying to lose weight” and “restricting carbs/fat/portions”). Agreement between coders was high (97%). Discrepancies were resolved via discussion between coders. Participants with comments coded as long-term diet changes were excluded in a sensitivity analysis (n = 24).

#### 2.2.4. Genotyping

Two buccal swabs (Whatman, OmniSwab) were collected from each participant (one from each cheek). Genomic DNA was isolated from the buccal swabs by the University of Arizona Genetics Core, quantitated, and used as the template in a TaqMan^®^ SNP Genotyping assay to determine the genotype at the rs738409 SNP, located in codon 148 of *PNPLA3*. Genotypes were determined for 100% of participants, and the minor allele (G) frequency was 51%. Risk allele carrier status was defined by *PNPLA3* rs738409 genotype, and individuals were categorized as CC, CG, and GG genotypes, corresponding to 0, 1, or 2 risk alleles, respectively.

#### 2.2.5. Statistical Procedures

Given a sample size of 288, an allele frequency of 0.5, an alpha of 0.05, a standardized beta of 0.26 for the genotype based on prior findings [33], and a standardized beta for the dietary variable of 0.5, we had >80% power to detect a significant interaction with a standardized beta of ~0.21 (*r^2^*~0.022). In addition, we believe that we are sufficiently powered based on a similar study examining *PNPLA3*-by-dietary sugar interactions among Hispanic children and adolescents (n = 153) in which significant interactions were discovered [19].

All continuous variables were examined visually for normality. Normally distributed variables were summarized as mean ± standard deviation and skewed variables were summarized as median and interquartile range (IQR). One-way analysis of variance was used to compare continuous variables between genotypes. If there were overall significant differences between genotype groups for any given continuous variable (*p* < 0.05), post-hoc pairwise comparisons with Bonferroni adjustments were used. Chi-squared tests were used to compare categorical variables between genotype groups. BMI had a non-linear relationship with the outcome of liver steatosis and was consequently log transformed to meet the assumption of linearity. Multiple linear regression was used to assess whether a diet x genotype interaction existed on the outcome of interest, liver steatosis. If the interaction term was not significantly associated with hepatic steatosis, it was subsequently removed from each model leaving the dietary variable and risk allele carrier status as independent predictors. An additive model was used such that genotypes were coded as 0, 1, or 2 which indicated the number of “G” risk alleles. The following dietary variables were selected based on significant findings in prior literature: total carbohydrate intake [19], total sugar intake [19], omega-3 PUFA intake [24], omega-6/omega-3 PUFAs [21]. Covariates were selected *a priori* based on the scientific literature and their association with liver steatosis. Covariates included in the model were sex, age, and BMI. Waist circumference was considered as a model covariate but was not included due to its high collinearity with BMI. All models were checked for the assumptions of linear regression: linearity, normality of residuals, and homoscedasticity. A significance level of *p* < 0.05 was set. All analyses were conducted in R Studio version 3.6.2 [34].

#### 2.2.6. Sensitivity Analyses

Given that participants were provided their liver scan results and informed that weight loss could attenuate fatty liver, a sensitivity analysis was conducted for all models that excluded participants with diet recall comments that reflected they were in the process of making dietary changes intended to produce weight loss (n = 24).

## 3. Results

Of the 778 individuals interested in participating in the study, 720 were screened and 316 were enrolled. Three hundred and seven participants (97%) had complete available demographic and clinical data from the cross-sectional study visit. Of these 307 participants, 288 (94%) completed at least two dietary recalls and therefore were included in the final sample. Specifically, nineteen participants were excluded for not completing at least two dietary recalls. A recruitment flow diagram outlining how the final sample of 288 participants was reached is summarized in Figure 1.

### Participant Characteristics

Full demographic and clinical characteristics of participants by *PNPLA3* genotype are summarized in Table 1. There were no significant differences in age, weight, BMI, waist circumference, liver fibrosis, income level, diabetes status, current use of lipid-lowering medication, nor insurance status between genotype groups. Mean liver steatosis values differed significantly across the three genotypes: (CC: 285.5 ± 53.4 dB/m; CG: 283.7 dB/m (44.5); GG: 302.4 dB/m (50.8)). Specifically, liver steatosis values were similar in CC and CG groups; but differed significantly between GG and CG groups.

Dietary variables by genotype for the final sample of 288 are summarized in Table 2. No significant differences in dietary variables were observed across genotype subgroups except for soluble fiber, wherein intake was lower in the GG versus the other two groups. Given the potential for diabetes status and/or use of lipid-lowering medication to influence dietary intake, we compared dietary intake between the participants who reported a current type 2 diabetes diagnosis and/or current use of lipid-lowering medication (n = 43) and those who did not (n = 245). No significant differences for any of the dietary variables were observed between the two groups except for % calories from added sugar (*p*-value = 0.04) (available in the Appendix A as Table A1).

In the primary analysis, no dietary variables were significantly associated with levels of hepatic steatosis (Table 3) and no statistically significant interactions were observed between *PNPLA3* genotype status and dietary intake in relation to levels of hepatic steatosis. Results from the sensitivity analysis excluding subjects who reported systematic dietary changes intended to produce weight loss were similar to the results from the primary analysis (available in the Appendix A as Table A2).

## 4. Discussion

We investigated interactions between *a priori*-selected dietary variables and *PNPLA3* rs738409 in relation to levels of hepatic steatosis assessed by transient elastography (Fibroscan^®^) in a sample of MO adults with overweight or obesity. This study did not identify any significant association between dietary intake and levels of hepatic steatosis nor any significant gene-diet interactions. To our knowledge, this was the first study to explore interactions between rs738409 (Ile148Met) and dietary intake in MO adults, a population disproportionately burdened by NAFLD with among the highest prevalence of this high-risk SNP.

Median liver steatosis values, as assessed by transient elastography, were lower among the men (287 dB/m) and higher among the women (293 dB/m) in our study than those of the men (306 dB/m) and women (277 dB/m) of a population-based sample of Mexican Americans living in Texas (n = 774) [35]. However, this is not surprising given the women in our sample had a greater median BMI than women in the Texas sample while the difference in median BMI among men in the studies was much smaller [35].

While our findings in MO adults compare favorably to previous studies in largely NHW populations that suggest limited support of an association between dietary intake of macronutrients and levels of hepatic steatosis, including total carbohydrates [36], sucrose [37], fructose [37,38], and total fat [36,39], a number of studies have reported significant associations between specific macronutrients and hepatic steatosis [40]. The lack of clarity in associations between specific macronutrient intake and NAFLD observed in the literature may be due to one or more of the following: (1) whether or not BMI was included as a covariate [36]; (2) differences in specific food sources of these nutrients (fructose derived from fruit vs. sweetened beverages) [36,40]; and (3) challenges in distinguishing the effects of specific nutrients from the effects of an overall high-caloric diet and diet quality [36,40,41].

Despite the lack of significant genotype x diet interactions observed here in MO adults, significant interactions between *PNPLA3* genotype status and total carbohydrate intake (% kcals/d) [19], total sugar intake (% kcals/d) [19], fructose-sweetened beverages [42], and omega-6/omega-3 PUFAs [21] have been observed in Hispanic pediatric populations. Interactions also have been explored between this SNP and dietary patterns in a cross-sectional sample of Italian children and adolescents with obesity [42]. In the study, it was found that the variant was more strongly associated with steatosis in those who reported consuming sweetened beverages at least once per week [42]. In the same study, a significant interaction was observed between the variant and intake of vegetable intake such that the association of the variant with steatosis was weaker in participants who consumed a diet low in vegetables [42].

In a cross-sectional analysis among adults that assessed interactions between *PNPLA3* genotype status and dietary intake (sucrose intake, carbohydrate intake, and omega-6/omega-3 PUFAs) in relation to serum triglycerides, investigators stratified the 4827 Swedish participants by BMI status (i.e., normal or overweight) in addition to genotype status (i.e., CC, CG, or GG) [25]. Interestingly, an interaction between rs738409 and sucrose intake in relation to serum triglycerides was only significant among normal-weight participants in the highest tertile of sucrose intake [25]. Our study only recruited individuals with overweight and obesity and this undermined our ability to evaluate interactions between *PNPLA3* genotype status and dietary intake in normal weight individuals. Additionally, the Swedish study found that the G-allele was associated with lower serum triglycerides only among overweight participants in the lowest tertile of carbohydrate and omega-6/omega-3 PUFA intake [25]. These data suggest that the magnitude and direction of the effect of the rs738409 variant on liver-related outcomes may depend on weight status and may be further modified by level of dietary intake of carbohydrate, sucrose, and omega-6/omega-3 PUFAs [25].

Another potential explanation as to why we did not observe any significant main effects or interactions may be related to the considerable underreporting of energy intake among our study participants across genotype groups. Our study participants reported consuming an average of 1496 ± 540 kcals per day (fixed mean for all groups combined) whereas the estimated average kcal intake for American adults in 2009–2010 with overweight or obesity was 2320.4 ± 32.4 and 2116.9 ± 20.5 kcals per day, respectively [43]. Individuals at the greatest risk of underreporting have consistently been found to be women with overweight, obesity, less education, and less income, all prevalent characteristics among our study participants and U.S. Hispanics/Latinos [44,45] more broadly. Despite the stringent dietary assessment methodology utilized in this study, including the use of three 24-h dietary recalls and the USDA Automated Multiple-Pass Method, the likely underreporting observed in this study may have limited our ability to detect gene-diet interactions, characterized by modest effect sizes, by contributing to greater statistical “noise” [46]. Additionally, the high level of underreporting may have influenced the effect size of dietary intake on levels of hepatic steatosis, contributing to reduced statistical power.

Additionally, while hepatic fat fraction increased in a dose-dependent fashion based on the number of risk alleles in Davis et al. (2010) [19] and Santoro et al. (2012) [21], this was not the case in our study, where levels of hepatic steatosis were slightly lower in adults carrying one risk allele (i.e., CG individuals) vs. those carrying none (i.e., CC individuals). Potential explanations for this observation could be related to variables that were not summarized in the current study previously shown to be associated with levels of hepatic steatosis, including measures of body composition, levels of physical activity, and recent weight gain or loss.

### 4.1. Strengths and Limitations

Strengths of the current study include our focus on MO adults who are underrepresented in the liver disease literature and have among the highest rates of obesity, NAFLD, and prevalence of the *PNPLA3* variant. Our stringent diet assessment methodology included the use of three 24-h dietary recalls performed by bilingual and bicultural research assistants. Additionally, our eligibility criteria helped to control for potentially confounding variables known to influence liver steatosis independent of diet.

This study is not without its limitations. Self-reported dietary assessment methods are subject to bias resulting from a loss of memory of foods consumed, conscious omission of foods, misjudgment of portion size, and social desirability. In an attempt to minimize error, we assessed caloric plausibility of dietary intake. Future research should consider more objective measures of energy exposure, such as doubly labeled water, although costs can make this assessment method prohibitive in large, population-based studies. Another study limitation was related to the timing of the recalls, which took place after the in-person study visit in which participants were informed of the relationship between weight and liver fat. Future studies should be designed so that dietary intake is assessed before these associations are shared with participants, particularly if participants will be receiving any sort of lifestyle recommendations as part of the study or during follow-up discussions with medical professionals. However, we did address this limitation by performing a sensitivity analysis to exclude any participants who reported active dietary changes intended to produce weight loss, which did not alter our study conclusions.

### 4.2. Implications of Research

The *PNPLA3* genotype holds the potential to inform on differential risk profiles and effective dietary intervention strategies that can be integrated into patient care. To our knowledge, we are the first to report that *PNPLA3* genotype may not modify the relationship between dietary intake and hepatic steatosis in MO adults with overweight or obesity. Our null findings imply that the benefits and/or deleterious effects of dietary intake on levels of hepatic steatosis may be the same for MO adults regardless of their level of genetic risk associated with *PNPLA3* genotype status. Given the dearth of research examining these gene by diet interactions in MO adults, future studies are needed to enhance and replicate our study. To address issues related to underreporting of caloric intake, future studies should consider the use of strategies shown to reduce measurement error including a blended approach of multiple 24-h recalls and a food frequency questionnaire, the use of portion-size estimation aids, and/or the use of biomarkers as a means to verify self-reported dietary intake. Biomarker calibrated self-reported dietary assessments have been found to reduce bias commonly associated with self-reported methods and more clearly elucidate associations between diet and disease risk [47]. The use of biomarkers for sugar, fructose, and PUFA intake would be particularly relevant in studies exploring associations between dietary intake and NAFLD. While some biomarkers for dietary sugar intake show promise, controlled feeding studies are needed to assess the validity, reliability, and sensitivity of these biomarkers for different study designs [48].

## 5. Conclusions

The field of precision nutrition is well on its way towards identifying key genetic variants that can be used to guide dietary intervention recommendations [49]. Nutritional genetics provides an opportunity to study the complex gene-diet interactions involved in individuals’ personal pathogenesis of NAFLD [50] and answer the question, “*Are there ways in which we can we leverage an individual’s ‘nature’ to ‘nurture’ them more effectively in the prevention and treatment of NAFLD?*” However, major challenges persist related to replicability of findings in diverse populations and imprecise assessment of dietary exposures [49]. Given the rapidly increasing rates of NAFLD in MO populations, future research should continue exploring interactions between NAFLD-related SNPs, found to significantly increase risk and severity of NAFLD, and dietary intake in this population. Understanding the complex interactions between genetics and diet and the relationship to NAFLD will advance the field of precision nutrition by supporting tailored lifestyle interventions based on individual genetic background. The need to focus specifically on MO populations, who represent over 60% of U.S. Hispanics, is warranted given it is projected that U.S. Hispanics will represent approximately 30% of the country’s population by 2050 [51].

## Figures and Tables

**Figure 1 ijerph-18-07055-f001:**
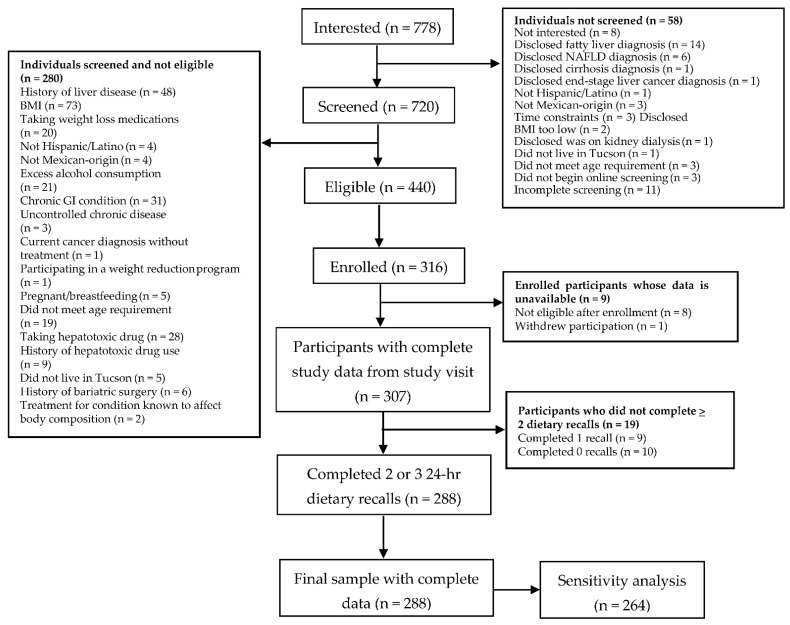
Cross-Sectional Study Participant Flow Diagram.

**Table 1 ijerph-18-07055-t001:** Demographic and Clinical Characteristics of Mexican-Origin Participants by *PNPLA3* Genotype (n = 288) ^1^.

Variable	*CC* (n = 72)	*CG* (n = 139)	*GG* (n = 77)	*p*-Value ^2^
Sex, n (%)				0.56
Male	23 (31.9)	54 (38.8)	26 (33.8)	
Female	49 (68.1)	85 (61.2)	51 (66.2)	
Age (y), mean ± SD	44.9 ± 12.1	44.2 ± 10.8	45.5 ± 10.8	0.69
Weight (kg), median (IQR)	86.6 (74.1, 99.5)	84.6 (77.6, 93.5)	87.5 (76.0, 100.2)	0.50
BMI (kg/m^2^), median (IQR)	31.3 (28.0, 35.8)	31.1 (28.9, 34.2)	32.6 (29.3, 36.3)	0.27
Waist Circumference (cm), median (IQR)	103.3 (97.0, 116.1)	103.0 (96.4, 111.1)	106.8 (99.2, 113.9)	0.18
Liver Steatosis (dB/m) ^3^, mean ± SD	285.5 ± 53.4	283.7 ± 44.5 ^a^	302.4 ± 50.8 ^b^	0.02
Liver Fibrosis (kPa) ^4^, median (IQR)	5.1 (4.1, 5.9)	5.0 (4.4, 6.2)	5.0 (4.5, 6.6)	0.50
Income Level, n (%)				0.18
<$29 K	35 (48.6)	61 (43.9)	47 (61.0)	
$30–59 K	24 (33.3)	55 (39.6)	21 (27.3)	
>$60 K	13 (18.1)	23 (16.5)	9 (11.7)	
Diabetes (yes), n	6 (8.3)	14 (10.1)	8 (10.4)	0.90
Insurance (yes), n	41 (56.9)	92 (66.2)	45 (58.4)	0.33
Current Use of Lipid-Lowering Medication (yes), n	5 (6.9)	9 (6.5)	6 (7.8)	0.93

^1^ All variables presented as either mean ± SD or median (IQR). Values with different superscripts are significantly different (*p*-value < 0.05). ^2^
*p* values represent overall significance between genotypes using ANOVAs and chi-square tests. ^3^ Possible CAP values for liver steatosis ranged from 100 to 400 dB/m. ^4^ Liver stiffness used to estimate levels of liver fibrosis; values range from 2 to 75 kPa.

**Table 2 ijerph-18-07055-t002:** Self-Reported Repeat 24-h Recall Dietary Intake of Mexican-Origin Participants by Genotype (n = 288) ^1^.

Variable	*CC* (n = 72)	*CG* (n = 139)	*GG* (n = 77)	*p*-Value ^2^
**Energy** (kcals)	1459 (1160,1682)	1426 (1166, 1822)	1430 (1079, 1730)	0.42
**Protein** (g/d)	66.4 (53.5, 78.0)	63.4 (49.4, 79.2)	61.9 (47.8, 73.7)	0.45
**Protein** (% of energy)	18.8 (14.9, 23.7)	17.4 (15.1, 20.8)	17.9 (14.6, 20.6)	0.24
**Fat** (g/d)	55.8 (40.4, 73.0)	56.3 (40.0, 74.0)	54.5 (39.4, 65.7)	0.57
**Fat** (% of energy)	33.8 (30.5, 37.8)	33.4 (30.0, 39.4)	36.6 (30.8, 39.0)	0.72
**n-3 PUFA** (g/d)	1.3 (0.9, 1.9)	1.3 (0.8, 2.0)	1.2 (0.8, 1.7)	0.514
**n-3 PUFA** (% of energy)	0.8 (0.6, 1.1)	0.8 (0.6, 1.1)	0.8 (0.6, 1.0)	0.55
**n-6 PUFA** (g/d)	11.2 (7.6, 15.9)	10.9 (7.9, 15.3)	10.8 (6.4, 14.0)	0.34
**n-6 PUFA** (% of energy)	6.7 (5.6, 8.5)	6.8 (5.8, 8.5)	6.6 (5.2, 7.6)	0.36
**n-6/n-3 PUFA**	8.3 (7.0, 9.2)	8.4 (7.5, 9.9)	8.3 (7.3, 10.0)	0.71
**Carbohydrates** (g/d)	175.4 (132.4, 208.3)	166.9 (131.9, 218.3)	169.0 (120.3, 211.1)	0.44
**Carbohydrates** (% of energy)	46.1 (42.0, 51.2)	46.7 (41.8, 51.3)	45.1 (39.4, 51.3)	0.65
**Fructose** (g/d)	13.6 (7.9, 21.2)	13.3 (7.8, 20.5)	13.3 (6.6, 20.9)	0.74
**Fructose** (% of energy)	3.8 (2.6, 5.6)	3.5 (2.5, 5.5)	3.7 (2.0, 5.8)	0.78
**Total sugar** (g/d)	59.4 (46.4, 85.3)	64.6 (45.7, 87.7)	61.7 (45.4, 84.6)	0.75
**Total sugar** (% of energy)	17.6 (13.2, 22.9)	17.6 (14.1, 23.1)	18.1 (13.8, 24.3)	0.84
**Added sugar** (g/d)	35.5 (19.7, 52.9)	39.2 (23.3, 57.8)	35.5 (22.3, 57.7)	0.73
**Added sugar** (% of energy)	9.3 (5.9, 13.5)	10.6 (7.0, 15.5)	10.3 (6.4, 16.1)	0.45
**Dietary fiber** (g/d)	16.8 (11.8, 22.2)	14.9 (11.0, 21.1)	13.7 (11.0, 18.6)	0.06
**Insoluble fiber** (g/d)	10.2 (8.1, 14.5)	9.6 (7.0, 13.6)	8.8 (6.9, 12.0)	0.12
**Soluble fiber** (g/d)	5.7 (4.5, 8.0)	5.1 (3.9, 7.5) ^a^	4.8 (3.5, 6.1) ^b^	0.02
**Glycemic index**	56.4 ± 4.9	56.7 ± 4.6	57.6 ± 5.1	0.30
**Glycemic load**	90.1 (66.3, 104.0)	87.2 (65.8, 110.7)	89.2 (62.3, 108.6)	0.62

^1^ All variables presented as median (IQR) except for glycemic index, presented as mean ± SD. Values with different superscripts are significantly different (*p*-value = 0.05). ^2^
*p* values represent overall significance between genotypes using ANOVAs and chi-square tests.

**Table 3 ijerph-18-07055-t003:** Coefficients for the associations of dietary variables of interest with liver steatosis from separate multivariable linear regression models.

	Coefficient	SE	*p*-Value
**Energy** (per 100 kcals)	−0.246	0.511	0.63
**Protein** (% of energy)	−0.136	0.475	0.78
**Fat** (% of energy)	0.629	0.391	0.11
**n-3 PUFA** (% of energy)	1.072	2.612	0.68
**n-6 PUFA** (% of energy)	−0.090	0.410	0.83
**n-6/n-3 PUFA**	−0.276	0.985	0.78
**Carbohydrates** (% of energy)	−0.391	0.310	0.21
**Fructose** (% of energy)	0.274	1.012	0.79
**Total sugar** (% of energy)	−0.220	0.369	0.55
**Added sugar** (% of energy)	−0.182	0.403	0.65
**Dietary fiber** (g/d)	0.197	0.362	0.59
**Insoluble fiber** (g/d)	0.524	0.523	0.32
**Soluble fiber** (g/d)	−0.475	0.980	0.63
**Glycemic index**	−0.678	0.546	0.22
**Glycemic load**	−0.082	0.068	0.23

**Notes:** All models controlled for sex, age, body mass index, and *PNPLA3* rs738409 risk allele carrier status.

## Data Availability

The data presented in this study are available on request from the corresponding author. The data are not publicly available due to the sensitive nature of the genetic information collected as part of the study.

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
