# Peer review of "Assessing Interactions between PNPLA3 and Dietary Intake on Liver Steatosis in Mexican-Origin Adults"

_ijerph, 2021, doi:10.3390/ijerph18137055_

Round 1

Reviewer 1 Report

I red with interest the manuscript by Morril et al. on the interaction between PNPLA3 polymorphism and dietary intake and their impact on hepatic steatosis in obese Mexican-origin adults.

Overall, it is a well written manuscript.

The element of novelty is the study population, characterized by an higher prevalence of obesity and by an higher prevalence of the PNPLA3 risk allele compared to other groups. My major concerns are related to the study design and to the statistical robustness.

Major revision:

  • The patients included in the study were obese and some of them had diabetes but in the manuscript there is no mention to the anti-hyperglycemic or lipids-lowering therapies. Pharmacological treatment could impact on diet as well as on hepatic steatosis.
  • There are no correlations between dietary parameters and PNPLA3 polymorphisms or hepatic steatosis. Why regression analysis were not adjusted for the PNPLA3 genotypes? I suggest to use the higher cut-off for CAP to perform a logistic regression analysis to assess the association between diet parameters and hepatic steatosis including in the model the PNPLA3 polymorphism.

Reviewer 2 Report

These investigators have recruited individuals who have identified themselves as having Mexican origin.  They examined diet intake and assessed liver steatosis and correlated the results against PNPLA3 polymorphism.  Comments:

  1. Do the allele frequency of this group match other Hispanic groups reported in the literature?  Does the frequency of the high risk polymorphism equate to the higher risk for steatosis among MO adults?
  2. In the end, these authors suggest that the diet has little interaction with the level of steatosis.  Does this allow one to conclude the argument of nature versus nuture, or genetics vs. environment?
  3. Are there other SNPs that have been reported that could be included in a subsequent analysis, given that they likely have the DNA already?

Round 2

Reviewer 1 Report

I would like to thank you the authors for improving the manuscript as suggested.